# Discovery of Novel Thiophene/Hydrazones: In Vitro and In Silico Studies against Pancreatic Cancer

**DOI:** 10.3390/pharmaceutics15051441

**Published:** 2023-05-09

**Authors:** Goknil Pelin Coskun, Yagmur Ozhan, Vladimir Dobričić, Jelena Bošković, Rengin Reis, Hande Sipahi, Zafer Sahin, Seref Demirayak

**Affiliations:** 1Department of Pharmaceutical Chemistry, Faculty of Pharmacy, Acibadem Mehmet Ali Aydinlar University, 34752 Istanbul, Turkey; 2Department of Pharmaceutical Toxicology, Faculty of Pharmacy, Yeditepe University, 34755 Istanbul, Turkey; yagmur.ozhan@yeditepe.edu.tr (Y.O.); hande.sipahi@yeditepe.edu.tr (H.S.); 3Department of Pharmaceutical Chemistry, Faculty of Pharmacy, University of Belgrade, 11221 Belgrade, Serbia; vladimir.dobricic@pharmacy.bg.ac.rs (V.D.);; 4Department of Pharmaceutical Toxicology, Faculty of Pharmacy, Acibadem Mehmet Ali Aydinlar University, 34752 Istanbul, Turkey; rengin.reis@acibadem.edu.tr; 5Department of Pharmaceutical Chemistry, Hamidiye Faculty of Pharmacy, University of Health Sciences, 34668 Istanbul, Turkey; zafer.sahin@sbu.edu.tr; 6Department of Pharmaceutical Chemistry, Faculty of Pharmacy, Kocaeli Health and Technology University, 41090 Kocaeli, Turkey; seref.demirayak@kocaelisaglik.edu.tr

**Keywords:** hydrazone, COX-2, pancreatic cancer, 5-LOX, 3D cell culture

## Abstract

Cancer is the disease with the highest mortality. Drug studies contribute to promising treatments; however there is an urgent need for selective drug candidates. Pancreatic cancer is difficult to treat and the cancer progresses rapidly. Unfortunately, current treatments are ineffective. In this study, ten new diarylthiophene-2-carbohydrazide derivatives were synthesized and evaluated for their pharmacological activity. The 2D and 3D anticancer activity studies suggested the compounds **7a**, **7d**, and **7f** were promising. Among these, **7f** (4.86 µM) showed the best 2D inhibitory activity against PaCa-2 cells. Compounds **7a**, **7d** and **7f** were also tested for their cytotoxic effects on healthy cell line but only compound **7d** showed selectivity. Compounds **7a**, **7d**, and **7f** showed the best 3D cell line inhibitory effect according to spheroid diameters. The compounds were screened for their COX-2 and 5-LOX inhibitory activity. For COX-2, the best IC_50_ value was observed for **7c** (10.13 µM) and all compounds showed significantly lower inhibition compared to standard. In the 5-LOX inhibition study, compounds **7a** (3.78 µM), **7c** (2.60 µM), **7e** (3.3 µM), and **7f** (2.94 µM) demonstrated influential activity compared to standard. Regarding molecular docking studies, binding mode of compounds **7c**, **7e**, and **7f** to the 5-LOX enzyme were non-redox or redox types, but not the iron-binding type. As dual inhibitors of 5-LOX and pancreatic cancer cell line, **7a** and **7f** were identified as the most promising compounds.

## 1. Introduction

Pancreatic cancer (PC) has become more prevalent over the past several years. It contributes to 5% of all cancer-related mortality and makes up approximately 2% of all cancers. Most patients have no overt symptoms until the disease has progressed to advanced pancreatic metastasis, when tumor cells are invasive. It is one of the most fatal malignant tumors, and early diagnosis is challenging. Even after aggressive treatment, most patients eventually relapse and the 5-year survival rate is only 2–9% [1]. In recent years, the cytosine arabinoside analog gemcitabine (2V, 2V-difluorodeoxycytidine) alone or in combination with other antineoplastic drugs, such as docetaxel and irinotecan, has displayed greater clinical activity over 5-fluorouracil, although survival was only marginally improved. PC was almost exclusively treated with fluoropyrimidines (e.g., fluorouracil) [2]. Pancreatic tumors are divided into two categories according to their nature of being either endocrine dependent or non-endocrine dependent. There are two types of non-endocrine pancreas tumors: benign and malignant. Different histological characteristics are present in malignant pancreatic tumors [3].

Prostaglandins are synthesized by a class of rate-limiting specific enzymes cyclooxygenases (COX), of which COX-2 is a prominent isozyme form in inflammatory processes [4]. COX-2 is ubiquitously expressed in a multitude of tumor forms, particularly pancreatic, prostate, and colorectal, which is correlated with a poor prognosis and aggressive cancer behavior [5]. Studies have shown that in pancreatic ductal adenocarcinomas, COX-2 expression is increased [6]. Compared to normal pancreatic tissue, pancreatic tumors show considerably higher COX-2 expression. It was found that in the pancreatic tumor cell line (BxPC-3) with the greatest levels of COX-2 expression occurred independently of Erk1/2 activation. Taken together, the literature data indicates that COX-2 may be a key factor in the development of pancreatic tumors and is thus a prospective chemotherapeutic target for the treatment of PC [7,8,9,10]. PC is associated with increased 5-lipoxygenase signalling (5-LOX) and inhibitors of 5-LOX enzyme may be effective in treating this disorder. According to epidemiological research, COX-2 inhibitors greatly reduced the incidence of colon, prostate, and breast cancer. Ding et al. investigated whether the COX-2 inhibitor Celecoxib and the 5-LOX inhibitor MK886 had any synergistic effect on the inhibition of PC cell proliferation because there was evidence that the combination of COX-2 and 5-LOX inhibitors had additive antitumor effect in colon cancer. According to the literature, combining the two compounds is an efficient strategy to halt the proliferation of the PC cell line SW1990 [5]. It is widely acknowledged that selective COX-2 inhibition causes the 5-LOX pathway to dominate leukotriene production and increases the prothrombotic effect by diverting the arachidonic acid metabolism. The significance of the arachidonic acid metabolism-related inflammatory responses in PC is thus supported by preclinical and clinical findings [11].

Additionally, because they have electron-rich and -poor sites, two nitrogen atoms, and a carbon atom, hydrazones [R_1_R_2_C = NNR_3_R_4_] are useful pharmacophores for the design of heterocyclic compounds with various biological features. As anti-inflammatory (AI) agents and specific COX and LOX inhibitors, many synthetic hydrazones have been tested [12]. Due to their extensive spectrum of pharmacological properties, hydrazones, a class of physiologically active chemical compounds of the Schiff base family, have piqued the interest of medicinal chemists. Many scientists are developing these substances as medicines to treat ailments while minimizing the side effects. There have been reports of several hydrazone derivatives having noticeable biological effects [13].

Therefore, in this study, we have synthesized ten new hydrazone compounds and investigated their anticancer properties using the PC cell line MIA PaCa-2 (2D and 3D systems). In addition, the COX-2 and 5-LOX inhibitory properties of these compounds were investigated.

## 2. Materials and Methods

### 2.1. General

All chemicals used were purchased from Merck (Darmstadt, Germany) and Sigma Aldrich (St. Louis, MO, USA). Reactions were monitored by thin layer chromatography (TLC) (petroleum ether/acetone; 7/3; *v*/*v*) on silica gel plates purchased from Merck (Merck). The melting points of the synthesized compounds were determined in a Stuart SMP50 Automatic Melting Point apparatus, and these are uncorrected. The purity of the synthesized compounds was confirmed by TLC and liquid chromatography–mass spectrometry (LC-MS). Nuclear magnetic resonance (NMR) spectra were recorded on a BRUKER 400 MHz (Billerica, MA, USA) for ^1^H-NMR and ^13^C-NMR. Data are reported as follows: chemical shift (ppm), multiplicity (b.s.: broad singlet, d: doublet; m: multiplet, s: singlet, and t: triplet), coupling constants (Hz), and integration. An Agilent 1260 Infinity II LC-MS equipped with a G7114A 1260DAD detector, G7311B 1260 Quad Pump system, G1328C 1260 manual injection unit, and G6125B LC/MSD detector were used for both LC and mass analysis. Retention times were recorded with an ACE C18 column (particle size: 3 µm, pore size: 100A). The column temperature was adjusted to 25 °C in the column compartment. The mobile phase consisted of an acetonitrile:water (80:20, *v*/*v*) mixture and delivered at a flow rate of 0.8 mL/min. The injection volume was 20 µL. The UV detector was operated at 254 nm. Rf × 100 values were recorded in petroleum ether/ethyl acetate/gl acetic acid at different ratios. 

### 2.2. Synthesis

Benzoin (**2**), 1,2-diphenylethan-1-one (**3**), 3-chloro-2,3-diphenylpropanal (**4**), and ethyl 4,5-diphenylthiophene-2-carboxylate (**5**) were synthesized according to a previously reported method [14]. 4,5-diphenylthiophene-2-carbohydrazide was synthesized via ester in the presence of hydrazine (Figure 1).

#### Hydrazone Synthesis

Substituted aldehydes (1 mmol) were added to hydrazide (1 mmol) in the presence of ethanol (20 mL) and the reaction was monitored by TLC. The reaction was heated under reflux for 2–6 h. The crude product was filtered and recrystallized by ethanol.


**N’-[(*E*)-(2-hydroxyphenyl)methylidene]-4,5-diphenylthiophene-2-carbohydrazide (7a)**


Yield: 67%; m.p.: 238 °C. ^1^H NMR (400 MHz, DMSO-*d*_6_, ppm) δ 12.25 − 12.11 (m, 1H, NHs), 11.15 − 10.98 (m, 1H), 8.65 (s, 2H), 8.08 (d, *J* = 1.2 Hz, 1H), 7.44 − 7.36 (m, 11H), 6.95 (dd, *J* = 7.8, 5.1 Hz, 1H). ^13^C NMR (101 MHz, DMSO-*d*_6_, ppm) δ 157.77 (C=O), 157.63, 147.90, 143.69, 138.90, 136.37, 135.66, 133.33, 132.33, 131.99, 129.44, 129.40, 129.34, 129.15, 129.06, 128.94, 128.08, 119.89, 119.35, 116.86.


**N’-[(*E*)-(3-hydroxyphenyl)methylidene]-4,5-diphenylthiophene-2-carbohydrazide (7b)**


Yield: 89%; m.p.: 225 °C. ^1^H NMR (400 MHz, DMSO-*d*_6_, ppm) δ 11.92 (s, 1H, NH), 9.70 (d, *J* = 11.2 Hz, 1H), 8.21 (m, 2H), 7.42 − 7.28 (m, 13H), 6.86 (d, *J* = 8.0, 2.4 Hz, 1H). ^13^C NMR (101 MHz, DMSO-*d*_6_, ppm) δ 161.38 (C=O), 158.20, 157.81, 148.19, 146.63, 145.14, 143.44, 138.84, 137.54, 136.93, 135.72, 133.39, 132.13, 130.51, 129.38, 128.05, 127.85, 119.41, 119.05, 118.04, 117.85, 113.90, 113.07.


**N’-[(*E*)-(4-hydroxyphenyl)methylidene]-4,5-diphenylthiophene-2-carbohydrazide (7c)**


Yield: 68%; m.p.: 256 °C. ^1^H NMR (400 MHz, DMSO-*d*_6_, ppm) δ 11.77 (s, 1H, NH), 9.99 (s, 1H), 8.51 − 7.84 (m, 2H), 7.61 (d, *J* = 9.2 Hz, 2H), 7.43 − 7.31 (m, 10H), 7.02 − 6.61 (m, 2H). ^13^C NMR (101 MHz, DMSO-*d*_6_, ppm) δ 168.00 (C=O), 161.11, 160.01, 157.57, 148.43, 146.42, 145.14, 143.09, 138.80, 137.34, 135.79, 133.72, 131.81, 129.42, 129.15, 128.98, 128.04, 125.60, 116.37, 116.23.


**N’-[(*E*)-(2-methoxyphenyl)methylidene]-4,5-diphenylthiophene-2-carbohydrazide (7d)**


Yield: 84%; m.p.: 263 °C. ^1^H NMR (400 MHz, DMSO-*d*_6_, ppm) δ 11.95 (s, 1H, NH), 9.20 − 8.36 (m, 1H), 8.10 (d, *J* = 2.9 Hz, 1H), 7.92 (dd, *J* = 20.1, 7.8 Hz, 2H), 7.44 − 7.29 (m, 10H), 7.13 (d, *J* = 8.4 Hz, 1H), 7.04 (t, *J* = 7.5 Hz, 1H), 3.87 (s, 3H). ^13^C NMR (101 MHz, DMSO-*d*_6_, ppm) δ 161.31 (C=O), 158.32, 157.71, 146.48, 143.41, 140.52, 138.82, 137.51, 137.05, 135.74, 133.69, 133.43, 132.13, 131.55, 129.37, 129.13, 127.82, 126.38, 125.96, 122.59, 121.27, 112.42, 112.31, 56.15 (OCH_3_).


**N’-[(*E*)-(3-methoxy-4-hydroxyphenyl)methylidene]-4,5-diphenylthiophene-2-carbohydrazide (7e)**


Yield: 75%; m.p.: 234 °C. ^1^H NMR (400 MHz, DMSO-*d*_6_, ppm) δ 11.81 (m, 1H, NH), 9.63 (s, 1H), 8.17 − 7.92 (m, 2H), 7.54 (d, *J* = 1.9 Hz, 1H), 7.36 (dt, *J* = 14.5, 5.1 Hz, 10H), 7.08 (dd, *J* = 20.6, 8.1 Hz, 1H), 6.85 (t, *J* = 9.1 Hz, 1H), 3.79 (s, 3H). ^13^C NMR (101 MHz, DMSO-*d*_6_, ppm) δ 161.08 (C=O), 157.63, 149.41, 148.63, 146.41, 144.76, 143.15, 138.81, 137.28, 135.76, 133.75, 131.58, 129.31, 129.13, 127.86, 125.98, 123.24, 122.81, 115.78, 109.35, 108.63, 55.50 (OCH_3_).


**N’-[(*E*)-(3-hydroxy-4-methoxyphenyl)methylidene]-4,5-diphenylthiophene-2-carbohydrazide (7f)**


Yield: 78%; m.p.: 249 °C. ^1^H NMR (400 MHz, DMSO-*d*_6_, ppm) δ 11.78 (d, *J* = 8.2 Hz, 1H, NH), 9.33 (d, *J* = 25.8 Hz, 1H), 8.46 − 7.85 (m, 2H), 7.41 − 7.28 (m, 11H), 7.11 (dd, *J* = 24.2, 8.4 Hz, 1H), 7.00 (t, *J* = 8.1 Hz, 1H), 3.82 (s, 3H). ^13^C NMR (101 MHz, DMSO-*d*_6_, ppm) δ 161.17 (C=O), 157.67, 150.38, 148.29, 147.39, 146.53, 145.20, 143.22, 138.80, 137.48, 137.17, 135.77, 133.72, 131.89, 129.37, 129.15, 128.03, 127.44, 120.93, 113.62, 112.75, 112.32, 56.09 (OCH_3_).


**N’-[(*E*)-(3,4-dimethoxyphenyl)methylidene]-4,5-diphenylthiophene-2-carbohydrazide (7g)**


Yield: 58%; m.p.: 253 °C. ^1^H NMR (400 MHz, DMSO-*d*_6_, ppm) δ 11.88 (s, 1H, NH), 8.11 − 8.02 (m, 2H), 7.56 (d, *J* = 1.9 Hz, 1H), 7.37 − 7.30 (m, 10H), 7.26 − 7.15 (m, 1H), 7.04 (t, *J* = 8.4 Hz, 1H), 3.86 − 3.64 (m, 6H). ^13^C NMR (101 MHz, DMSO-*d*_6_, ppm) δ 161.16 (C=O), 151.12, 149.55, 148.38, 146.47, 144.43, 137.30, 135.72, 133.64, 133.40, 131.93, 131.44, 129.30, 129.12, 128.07, 127.87, 127.26, 122.93, 122.56, 111.85, 108.61, 108.04, 56.03 (OCH_3_), 55.44 (OCH_3_).


**N’-[(*E*)-(2,3,4-trimethoxyphenyl)methylidene]-4,5-diphenylthiophene-2-carbohydrazide (7h)**


Yield: 87%; m.p.: 245 °C. ^1^H NMR (400 MHz, DMSO-*d*_6_, ppm) δ 12.15 − 11.58 (m, 1H, NH), 8.83 − 8.26 (m, 1H), 8.16 − 7.99 (m, 1H), 7.81 − 7.54 (m, 1H), 7.40 − 7.28 (m, 10H), 7.11 − 6.84 (m, 1H), 3.90 − 3.73 (m, 9H). ^13^C NMR (101 MHz, DMSO-*d*_6_, ppm) δ 161.11 (C=O), 157.60, 155.69, 153.21, 146.48, 143.52, 143.27, 142.08, 140.68, 138.82, 137.49, 137.10, 135.73, 133.71, 133.42, 131.96, 131.50, 129.39, 129.14, 128.03, 121.20, 120.62, 109.50, 62.30 (OCH_3_), 60.95 (OCH_3_), 56.50 (OCH_3_).


**N’-[(*E*)-(3,4,5-trimethoxyphenyl)methylidene]-4,5-diphenylthiophene-2-carbohydrazide (7i)**


Yield: 62%; m.p.: 254 °C. ^1^H NMR (400 MHz, DMSO-*d*_6_, ppm) δ 12.00 (s, 1H, NH), 8.21 − 7.86 (m, 2H), 7.37 − 7.28 (m, 10H), 7.21 − 7.01 (m, 2H), 4.32 − 3.53 (m, 9H). ^13^C NMR (101 MHz, DMSO-*d*_6_, ppm) δ 161.31 (C=O), 153.64, 148.19, 146.57, 144.09, 139.42, 137.50, 137.32, 135.70, 133.58, 132.10, 131.31, 130.07, 129.28, 129.22, 129.15, 129.12, 128.97, 128.07, 127.89, 104.83, 60.58, 56.43, 56.15 (OCH_3_).


**N’-[(*E*)-(2H-1,3-benzodioxol-5-yl)methylidene]-4,5-diphenylthiophene-2-carbohydrazide (7j)**


Yield: 73%; m.p.: 262 °C. ^1^H NMR (400 MHz, DMSO-*d*_6_, ppm) δ 11.87 (m, 1H, NH), 8.10 − 8.02 (m, 1H), 7.41 − 7.25 (m, 11H), 7.19 (d, *J* = 8.1 Hz, 1H), 7.15 (s, 1H), 7.01 (d, *J* = 7.9 Hz, 1H), 6.09 (d, *J* = 8.6 Hz, 2H). ^13^C NMR (101 MHz, DMSO-*d*_6_, ppm) δ 161.19 (C=O), 157.71, 149.67, 148.50, 147.92, 146.43, 144.58, 143.32, 138.82, 137.44, 137.03, 135.70, 133.68, 133.40, 131.98, 131.35, 129.39, 129.13, 128.04, 127.84, 124.18, 123.95, 109.07, 105.58, 102.08 (O-CH_2_-O).

### 2.3. Cell Culture Studies

For the preliminary assessment of anticancer activity in a 2D cell culture system, an MIA PaCa-2 human pancreas epithelial carcinoma (CRL-1420™) cell line (ATCC, Manassas, VA, USA) was cultured in Dulbecco’s Modified Eagle Medium with 10% fetal bovine serum (FBS, *v*/*v*) (Gibco, Billings, MT, USA) and supplemented with 1% antibiotics (10,000 μg/mL streptomycin and 10,000 U/mL penicillin) (Gibco, Billings, MT, USA) at 37 °C in a humidified 5% CO_2_ atmosphere. For the in vitro modeling of the tumor microenvironment, compounds were also tested in a 3D spheroid test system. For this purpose, the MIA PaCa-2 human pancreas epithelial carcinoma (CRL-1420™) cell line and human dermal fibroblast cell line HDF (PCS-201-012) cells were used. 

#### 2.3.1. 2D Anticancer Activity

For the assessment of anticancer activity of the compounds, the MIA PaCa-2 cells were plated in 48-well microplates at 1.5 × 10^5^ cells/mL and incubated for 24 h at 37 °C in 5% CO_2_ in accordance with a previously reported method [15]. Following the day, the cells were treated with compounds in various concentrations between 0.1–100 µM dissolved in dimethyl sulfoxide (DMSO) (Sigma Aldrich, St. Louis, MO, USA). Following the 72 h of incubation, culture mediums were discarded and 0.5 mg/mL 3-(4,5-Dimethylthiazol-2-yl)-2,5-Diphenyltetrazolium Bromide (MTT) (Sigma Aldrich, St. Louis, MO, USA) was added to each well for an additional 2 h at 37 °C. After 2 h incubation, the medium was discarded and 100 µL of isopropanol (Sigma Aldrich, St. Louis, MO, USA) was added to the wells to dissolve formazan crystals. The optical density was measured at 570 nm wavelength using a microplate reader (Thermo Fisher Scientific Multiskan, Finland). Doxorubicin hydrochloride (DOX) (European Pharmacopoeia standard, Sigma) was used as internal positive control, and 0.5% DMSO (*v*/*v*) was used as negative control. The percentage of cell viability was determined by using the following equation:(1)$$Viability\;\%=(Absorbance_{compound})/(Absorbance_{negative\;control})\times 100\%$$

#### 2.3.2. Cell Viability

HaCaT human healthy keratinocyte cells (CRL-2404™) (ATCC, Manassas, VA, USA) were cultured in DMEM (10% FBS, *v*/*v*) (Gibco, Billings, MT, USA), respectively and supplemented with 1% antibiotics (10.000 μg/mL streptomycin and 10.000 units/mL penicillin) (Gibco, Billings, MT, USA) at 37 °C in a humidified atmosphere of 5% CO_2_. For the preliminary assessment of the cytotoxicity of selected potent compounds in healthy cells, HaCaT cells were seeded 2 × 10^5^ cells/mL in 96-well plate and incubated for 24 h. The following day the **7a**, **7d**, and **7f** compounds were applied similarly as in the MIA PaCa-2 cytotoxicity protocol between concentrations 0.62–100 µM for 72 h and then MTT was applied.

#### 2.3.3. 3D Spheroid Formation/Growth Assay

For 3D spheroid formation, the MIA PaCa-2 cells (PC cell line) and HDF cells were used according to the 3D Bioprinting method [16]. To better simulate the tumor microenvironment, PC cells were mixed (1:1) with human fibroblasts. Briefly, the MIA PaCa-2 cells and human fibroblasts at 70–80% confluency were added to a 6-well plate and incubated with biocompatible NanoShuttle TM-PL (Bioprinting Kit, Greiner Bio-One, Germany) overnight at 37 °C in a humidified 5% CO_2_ atmosphere. After the nanoparticles were taken up by the cells, they were resuspended and seeded into an ultra-low attachment 96-well plate at a volume of 100 μL (2000 PC cells and 2000 human fibroblasts per well) volume. The plate was placed on a magnetic drive and incubated at 37 °C in a humidified 5% CO_2_ atmosphere until spheroids were formed. After two days of incubation, the spheroids were photographed using light microscopy (Carl Zeiss Primo Vert, Germany). Fresh medium containing 0.1, 1, 10, and 100 µM of tested compound was added to the wells. Photos were taken every 72 h, and the medium was replaced every 72 h. The effect of the tested compounds in 3D PC cell cultures was examined by measuring the size change of spheroids using ImageJ software (ImageJ 2.0 software, USA).

### 2.4. Enzyme Inhibition Assay

COX-1 and COX-2 inhibitory activities were tested using fluorometric COX-1 and COX-2 inhibitor screening kits (Abcam, United Kingdom), and the 5-LOX inhibitory activity was tested using a fluorometric 5-LOX inhibitor screening kit (MyBioSource, USA). These assays are based on the fluorometric detection of Prostaglandin G2 (a product generated by the COX enzyme) or on the fluorometric detection of the intermediate generated by the 5-LOX enzyme. The experiments were conducted in accordance with the manufacturers’ instructions [17,18,19].

For the determination of COX-1 and COX-2 inhibitory activity, the tested compounds were dissolved in dimethyl sulfoxide (DMSO, 5 mM stock solution) and then diluted with the same solvent to obtain the test solutions. The test solutions were further diluted 5× with COX Assay Buffer (2 µL of test solution: 8 µL COX Assay Buffer). Inhibitor Control (IC) was prepared by adding 2 µL of COX-1 (SC560) or COX-2 (Celecoxib) inhibitor and 8 µL COX Assay Buffer into the corresponding wells. Solvent Control (SC) was prepared by adding 2 µL DMSO and 8 µL COX Assay Buffer, while Sample (S) and Enzyme Control (EC) were prepared by adding 10 µL of diluted test solution and COX Assay Buffer, respectively, into assigned wells. 

For the determination of 5-LOX inhibitory activity, tested compounds were dissolved in DMSO (5 mM stock solution) and then diluted with the same solvent to obtain the test solutions. Test solutions were added to wells in an amount of 2 µL in the final 100 µL reaction volume. The EC, SC, and IC were prepared by adding 2 µL of Assay Buffer, DMSO, and Zileuton into the corresponding wells. Fluorescence of the samples were kinetically measured at 25 °C (COX-1 and COX-2: Ex/Em = 535/587 nm, for 10 min; 5-LOX: Ex/Em = 500/536 nm for 20 min) using Synergy LX multi-mode microplate reader (BioTek, USA). EC, SC, IC, and all test solutions were analyzed in triplicate.

### 2.5. Molecular Docking Study

For the investigation of the interactions of **7c**, **7e**, and **7f** with COX-2 and 5-LOX enzymes, the following crystal structures of: 5KIR (inhibitor Rofecoxib bound to COX-2) and 6N2W (inhibitor NDGA bound to 5-LOX) were downloaded from the Protein Data Bank [20]. The receptor sites were prepared using MAKE Receptor 3.2.0.2 software [21]. The molecular docking experiments were performed in the boxes where the dimensions were 21.67 Å × 20.67 Å × 23.67 Å (COX-2), and 19.67 Å × 14.67 Å × 22.67 Å (5-LOX) and the boxes were generated around the co-crystalized ligands. The outer contour sizes were 1751 Å^3^ (COX-2) and 908 Å^3^ (5-LOX). The setup of contours was set as “Balanced”, and no constraints were applied.

Prior to molecular docking, ligand preparation was performed in OMEGA 2.5.1.4 and files containing conformers of each ligand were generated. The OEDocking 3.2.0.2 software [22,23,24], which employs FRED (fast exhaustive docking) tool, was used for the analysis of ligand binding positions into the defined receptor sites of COX-2 and 5-LOX enzymes. Exhaustive scoring was performed using the Chemgauss4 scoring function. Further optimization was performed using the OEChemscore scoring function. Scoring and consensus position selection was performed using the Chemgauss4 scoring function. Other settings were set as default.

The docking validation was performed by docking of the structures of co-crystalized ligands into the active sites of both enzymes. The binding positions were compared with the conformations of co-crystalized ligands and the root-mean-square deviation (RMSD) values were calculated. In both cases, the RMSD values were <2.0 Å, which proves the validity of these in silico experiments [25]. 

## 3. Results and Discussion

### 3.1. Chemistry

The conventional benzoin condensation method was used as a starter of this study’s synthesis. Benzaldehyde was added to a solution of sodium cyanate and the reaction was performed in ethanol. The obtained benzoin was then reduced to the corresponding carbonyl derivative in the presence of tin and hydrochloric acid. The keto-enol tautomerism of 1,2-diphenylethan-1-one was allowed to compound to further the Vilsmeier reaction with additional chlorine substitution at the alpha position of the aldehyde, in the presence of phosphorous(V) oxychloride. The obtained 3-chloro-2,3-diphenylpropanal was converted to ethyl 3,4-diphenylthiophene-2-carboxylate in the presence of ethyl mercaptoacetate. The following reaction consisted of hydrazine hydrate addition to an ester derivative. The obtained hydrazide intermediate was converted to its corresponding hydrazone derivative by the addition of substituted benzaldehyde derivatives to generate the novel hydrazone-thiophen compounds (Figure 1). The yields of the compounds ranged 58%–89%.

The structures of the original hydrazone derivatives (**7a–f**) were elucidated by spectroscopic methods. The proton and carbon NMR analyses confirmed the structures of the synthesized compounds. The important outcome of hydrazone formation is the disappearance of hydrazide protons in ^1^H NMR [26] Instead of hydrazide protons, a new azomethine proton must be formed. As all the compounds were analyzed in both proton and carbon NMR, the azomethine proton and azomethine carbon peaks were recorded as expected. The azomethine protons were observed between 11.15 and 7.85 ppm and the azomethine carbons were observed between 160.01 and 149.41 ppm as expected.

All compounds were also examined for their purity and mass analysis in an LC-MS system. All compounds were observed as single peaks in high-performance liquid chromatography chromatograms and the molecular ion peaks were observed in the positive scan mode.

### 3.2. 2D Anticancer Activity

Anticancer activity of compounds was expressed as MTT-based IC_50_ values after 72 h of exposure to MIA PaCa-2 cell line. As represented in Table 1, the compounds **7e**, **7h**, and **7i** exhibited IC_50_ values > 100 µM. On the other hand, the lowest IC_50_ was observed for compound **7f** (4.86 ± 0.19 µM), which was the most potent derivative. The positive control used in this study was Doxorubicin.

According to the MTT assay results of **7a, 7d,** and **7f** in HaCaT cells, it was seen that **7d** exhibited an IC_50_ value above the highest tested concentration, 100 µM. On the other hand, **7a** and **7f** showed a higher cytotoxicity profile compared to **7d**. The IC_50_ values were given in Table 2.

### 3.3. Anticancer Activity in 3D Cell Cultures (Spheroids)

3D cell cultures have been widely utilized as a model to evaluate the anticancer properties of new chemicals. When compared to the traditional cell monolayers (2D models), this model is thought to better capture the true tumor microenvironment [27]. The effects of the tested compounds at different concentrations (0.1, 1, 10, and 100 µM) on MIA PaCa-2 cell spheroid growth were evaluated (Figure 2). The activity of compounds in 3D cell cultures was determined as the change in size of the spheroids (Table 3). After 9 days of treatment, spheroid size of the control group increased by almost 6.52 times. Compared to the control, at 10 µM DOX as a reference molecule showed an considerable decrease in spheroid size by 24%, after 9 days of treatment. As seen in Figure 2, no reduction in spheroid size was observed at the lowest doses (0.1 µM) of the tested compounds. At the 1 µM dose of the tested substances, **7a** showed the greatest spheroid size reduction. Spheroids incubated with **7e** (10–100 µM), **7f** (100 µM), **7g** (10–100 µM), and **7j** (100 µM) had disintegrated at the end of the 9 days. Disintegrated concentrations of compounds were excluded from the calculation in Table 2. As seen in Figure 2 and Table 3, at 10 µM dose, **7a** showed the greatest anticancer activity with a 46% reduction in tumor size growth. Moreover, after incubation with 100 µM of these compounds for 9 days, the spheroid diameter treated with **7a** indicated a reduction of 65% followed by 55% reduction in the **7d** treated group. In addition, **7a** and **7d** had substantially better anticancer properties than the known anticancer drug DOX. According to our results, the most active compound against MIA PaCa-2 tumor spheroids was **7a**. The active compounds on MIA PaCa-2 cells were also tested against healthy cell line for selectivity. According to the results presented in Table 3, it can be assumed that only compound **7d** showed selectivity on cancer studies. Compounds **7a** and **7f** were found to show cytotoxic effects on healthy cell line too.

Considering the substitutions of the synthesized compounds, the anticancer activity and the structure-activity relationship (SAR) can be assumed. Since the targeted compounds were chosen to have electron donating groups, the positions on the aromatic ring gave significant results with respect to SAR. 2D cell culture studies showed that compounds **7a, 7d**, and **7f** had the best IC_50_ values. From the cell culture results, it is clear that electron donating groups seem to favor anticancer activity. Compound **7f** exhibited the best IC_50_ value, which also supports the importance of electron donating groups. Compounds having only a hydroxyl and methoxy group in their para position showed lower activity. Hydroxyl or methoxy substitution in ortho position (compounds **7a** and **7d**) resulted in lower IC_50_ values. When the electron character of the structure is increased, as in compound **7f**, the best anticancer activity was observed. Among the tested compounds, compound **7j**, which has a benzodioxole ring, also showed a low IC_50_ value, which also supports the idea of an electron-rich character of the compounds. Compounds **7h** and **7i** showed a different perspective on SAR, as they both have three methoxy groups and both showed high IC_50_ values. This could be the result of steric hindrance of the methoxy groups in the compounds. The cell culture studies may indicate that compounds with electron donating groups in the ortho position have a good activity profile. When the electronic environment is increased, the biological activity leads to a lower IC_50_ value.

### 3.4. Enzyme Inhibition Results

The importance of the interplay between inflammation and cancer is well known. The most important inflammation mediators are prostaglandins and leukotrienes, which are produced by the COX-2 and 5-LOX enzymes (COX-1 produces physiologically important mediators and its inhibition is unwanted) [28,29,30]. 

The results of the inhibition of COX-2, and 5-LOX enzymes are presented in Table 4 showing the IC_50_ values and percent of inhibition at 10 µM. In addition COX-1 inhibition was investigated only for compounds with the greatest COX-2 inhibitory activity (**7c**, **7e**, and **7f**) and IC_50_ values were presented. For comparison, commercially available selective COX-2 and 5-LOX inhibitors, Celecoxib and Zileuton, respectively, were tested.

The best COX-2 inhibitors were **7c**, **7e**, and **7f**, which had IC_50_ values of <40 µM and the percent of inhibition at 10 µM was >38%. These three compounds also displayed weak COX-1 inhibitory activity, indicating their good COX-2/COX-1 selectivity. IC_50_ values of these compounds were higher, while their COX-2 percent of inhibition at 10 µM were lower compared to Celecoxib.

The best 5-LOX inhibitors were **7a**, **7c**, **7e** and **7f**, which had IC_50_ values of <10 µM and the percent of inhibition at 10 µM was >55%. Although these IC_50_ values were higher than the IC_50_ value of Zileuton, 5-LOX percent of inhibition at 10 µM of these compounds were similar to Zileuton. Therefore, **7b** could also be considered a 5-LOX inhibitor, but with weak inhibitory activity (IC_50_ = 93.23 ± 9.89).

The best dual COX-2 and 5-LOX inhibitors were **7c**, **7e**, and **7f** and these compounds have the greatest AI potential. On the other hand, only **7f** has remarkable cytotoxicity, indicating that the impact of the inhibition of these two enzymes on the MIA PaCa-2 cells viability is low and emphasizes the importance of other anticancer mechanisms. 

The enzyme inhibition studies for both COX and 5-LOX yielded an understanding of the mechanism of action that correlates better with the cell culture studies. It is clear from the study that the electron rich environment of the compounds favors the enzyme inhibitory activity. Molecular docking studies are also in agreement with the enzyme inhibiton results as substituted benzene ring forms π-alkyl or π-cation interactions with COX-2 and 5-LOX. It can be assumed that electron-donating groups could strengthen these interactions. The best IC_50_ value of COX -2 inhibition was observed for compound **7c**, which has only one hydroxyl group in its para position. When a methoxy group is added in the meta position, the activity decreases (which is also not a dramatic decrease). However, when the substituents were arranged differently, as in compound **7f**, the COX-2 inhibitory activity decreased. This could most likely be due to steric hindrance of the methoxy group when it is in para position. The activity profile of 5-LOX shows that the meta position, when alone, does not favor the inhibitory activity of 5-LOX (as in compound **7b**). The hydroxyl group alone showed good inhibition, while the hydroxyl and methoxy groups together also showed significant inhibition. When the substitutions at the para positions are not bulky groups, the activity of 5-LOX is high. Substitutions at the meta position do not inversely affect the activity, but actually favor the inhibition of 5- LOX. Dual inhibition for both COX-2 and 5-LOX was observed for compounds **7c**, **7e**, and **7f**, supporting the idea that an electron-rich environment is a key point for enzyme inhibition.

### 3.5. Molecular Docking Study

The analysis of binding of the tested compounds with the best COX-2 and 5-LOX inhibitory activity (**7c**, **7e** and **7f**) to these enzymes was investigated in silico using molecular docking. The binding results of **7c**, **7e**, and **7f** were compared to the binding results of selective COX-2 (Rofecoxib and Celecoxib) and 5-LOX inhibitors (Zileuton and NDGA).

#### 3.5.1. Molecular Docking into COX-2

Co-crystalized ligand Rofecoxib, which is a selective COX-2 inhibitor, interacted with the receptor in two different modes. The first one was characterized by π-alkyl interactions of benzene rings with LEU352, VAL523, and ALA527, hydrogen bond between the oxo groups and TYR355 as well as the hydrogen bond between the sulfonyl group and TYR385 (Figure 3a). The second mode was characterized by π-alkyl interactions of benzene rings with VAL349, LEU352, VAL523, and ALA527, as well as the hydrogen bonds between the sulfonyl group and ILE517 and PHE518 (Figure 3b).

Another selective COX-2 inhibitor Celecoxib formed π-alkyl or π-sigma interactions with VAL523, SER353, ALA527, VAL349, TYR355, LEU531, and LEU359, as well as hydrogen bonds between the sulfonamide group and ILE517, PHE518, LEU352, SER353, and GLN192 (Figure 4).

Thiophene and benzene rings of the tested compounds also formed π-alkyl interactions with VAL349, LEU352, VAL523, and ALA527, while the benzene ring on the opposite side of the molecule formed additional π-alkyl interactions with LEU93 and VAL116. The acyl hydrazone group formed a hydrogen bond with TYR355 (Figure 5). 

However, docking scores of the tested compounds (**7c**: −10.10 kcal/mol; **7e**: −9.73 kcal/mol; and **7f**: −8.75 kcal/mol) were higher than the docking scores of selective the COX-2 inhibitors Rofecoxib (−12.61 kcal/mol) and Celecoxib (−13.69 kcal/mol), which indicates a more favorable binding of Rofecoxib and Celecoxib to this enzyme and can explain their better in vitro activity. The better binding results of Rofecoxib and Celecoxib could be explained by the fewer number of important interactions, as well as the presence of sterically unfavorable interactions between the tested compounds and certain amino acids at the receptor site (ILE112 and LEU352). The latter is a result of the higher voluminosity of the tested compounds in comparison to Rofecoxib and Celecoxib. 

#### 3.5.2. Molecular Docking into 5-LOX

The 5-LOX inhibitors could be classified into three groups: iron-ligand, redox, and non-redox inhibitors. Iron-ligand inhibitors, such as Zileuton, inhibit the enzyme through the chelation of the iron atom and/or by stabilizing its ferrous state. Non-redox- and redox-type inhibitors, such as NDGA, compete with the substrate arachidonic acid for binding to the active site [31].

The co-crystalized ligand NDGA formed π-alkyl interactions with ASN407, ARG596, and ILE673, as well as hydrogen bonds with ASN407, ARG596, and ILE673. In addition, no interactions with iron were observed, which is expected since NDGA is redox-type inhibitor [32]. Molecular docking was also performed on Zileuton, an iron-ligand inhibitor. Our results showed that there is an interaction between Zileuton and iron, which confirms the validity of the in silico study presented in this paper (Figure 6).

The tested compounds did not interact with iron, which indicates that these compounds are redox or non-redox inhibitors. The thiophene and benzene rings of these compounds formed π-alkyl and π-sigma interactions with the hydrophobic pocket of the receptor, consisting of LEU368, ILE406, ALA410, and ARG411, while the benzene rings and its substituents on the opposite side of the molecule formed a hydrogen bond or π-cation interaction with ARG596. It can be noticed that all compounds that show 5-LOX inhibitory activity (**7a**, **7b**, **7c**, **7e**, and **7f**, Table 4) have at least one –OH group on the benzene ring, indicating the importance of this interaction within the 5-LOX active site. In addition, the acyl hydrazone groups of **7c** and **7f** formed hydrogen bonds with GLN363 (Figure 7). Docking scores of tested compounds (**7c**: −6.17 kcal/mol; **7e**: −6.09 kcal/mol; and **7f**: −5.98 kcal/mol) were higher than the docking scores of NDGA (−10.67 kcal/mol) and Zileuton (−8.77 kcal/mol), which indicates a more favorable binding of Zileuton and NDGA to this enzyme. 

## 4. Conclusions

In this study, ten new hydrazone-thiophen compounds were designed, synthesized, and their structures were elucidated using spectroscopic methods. The compounds were tested for their anticancer activity, and for COX-2 and 5-LOX inhibitory activity. The results revealed promising results for the tested molecules. This study represented a high point for a better understanding of anticancer activity and mechanism of action. One of the ideas of this study was to investigate the relationship between COX-2/5-LOX inhibition and pancreatic cancer. Among the compounds tested, compound **7f** showed promising results supporting the inhibitory profile of pancreatic cancer cell lines. However, this compound was found to have cytotoxic effects on healthy cells. Only compound **7d** showed selectivity on cancer studies. An electron- rich environment was shown to be beneficial for both anticancer activity and dual enzyme inhibition. The electron donating effect of the methoxy and hydroxyl groups increased the profile of anticancer activity. When the substitutions are in para and meta positions, the activity of the tested compounds is high. Our study also suggests that the synthesized compounds, which are dual inhibitors of 5-LOX and COX-2, may have great potential in terms of anticancer activity. Furthermore, the chemistry of the compounds also revealed the importance of electron donating substitution. This study could provide a basis for the development of new anticancer molecules against PC.

## Figures and Tables

**Figure 1 pharmaceutics-15-01441-f001:**
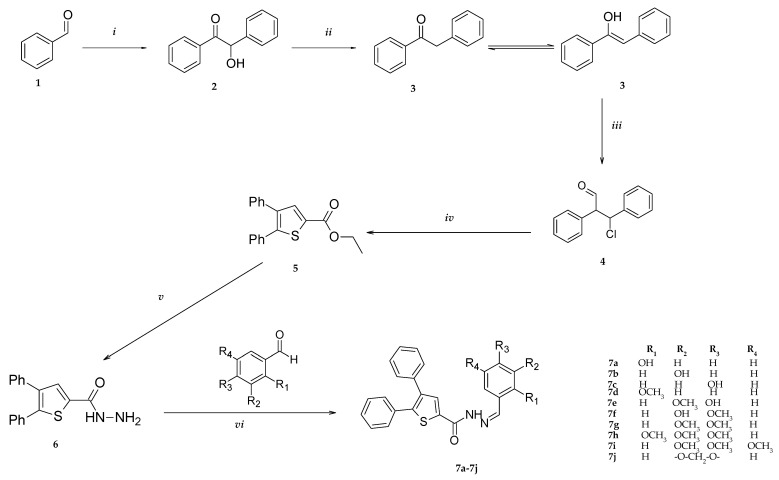
Synthesis of the compounds **7a–j**. Processes ***i***: NaCN, EtOH, reflux, 2h; ***ii***: Sn/HCl; ***iii***: N,N-DMF, POCl_3_,Vilsmeier; ***iv:*** ethyl mercaptoacetate, pyridine, TEA (trimethylamine) ***v***: Hydrazine hydrate, EtOH, reflux; and ***vi:*** appropriate aldehyde, ethanol.

**Figure 2 pharmaceutics-15-01441-f002:**
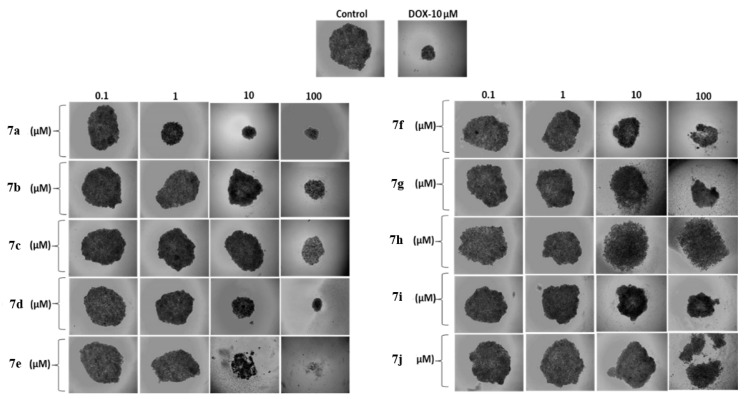
Effect of the tested compounds in 3D cell cultures. Photos of human pancreatic carcinoma MIA PaCa-2 tumor spheroids at the end of the experiment (after 9 days of incubation with different concentrations of compounds). DOX: Doxorubicin HCl. Magnification 4×.

**Figure 3 pharmaceutics-15-01441-f003:**
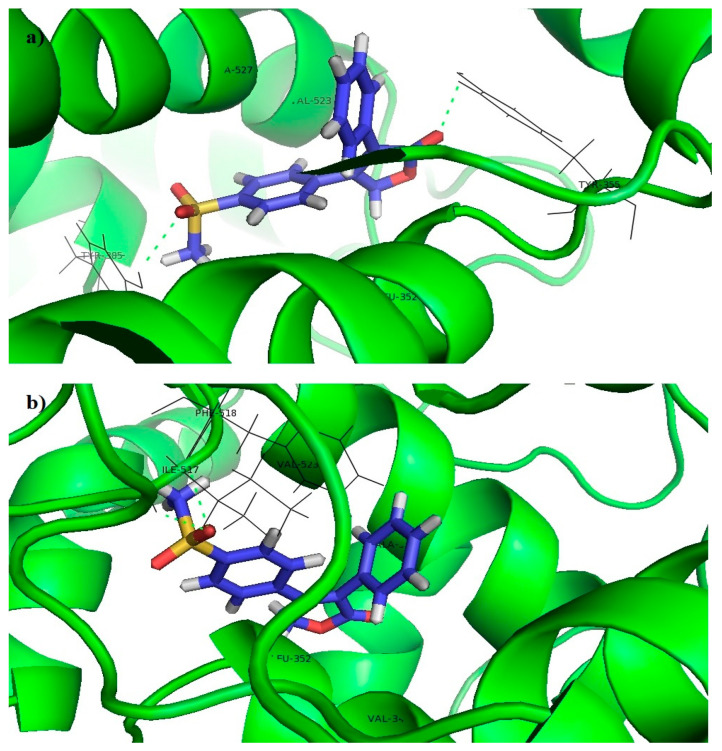
Molecular docking of Rofecoxib into the COX-2 enzyme: (**a**) First binding mode. (**b**) Second binding mode.

**Figure 4 pharmaceutics-15-01441-f004:**
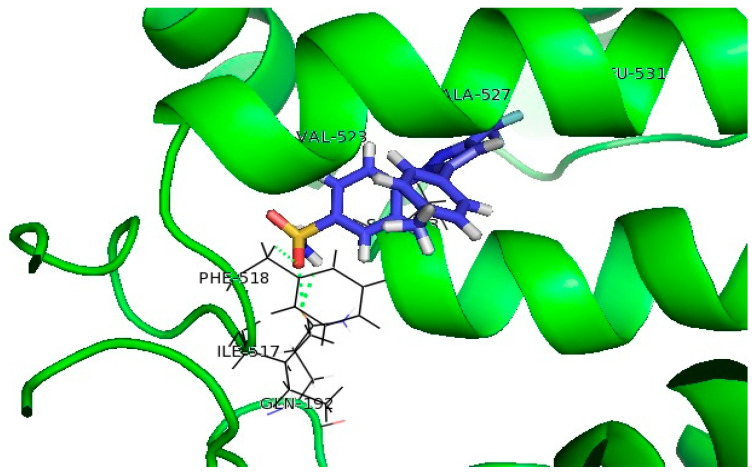
Molecular docking of Celecoxib into the COX-2 enzyme.

**Figure 5 pharmaceutics-15-01441-f005:**
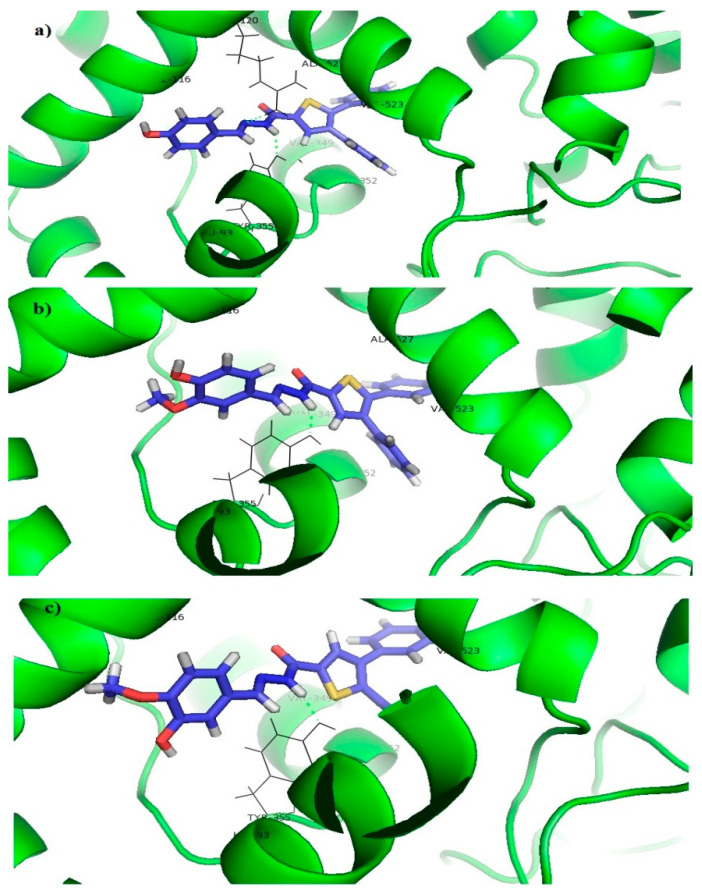
Molecular docking of tested compounds into the COX-2 enzyme: (**a**) **7c**, (**b**) **7e**, (**c**) **7f**.

**Figure 6 pharmaceutics-15-01441-f006:**
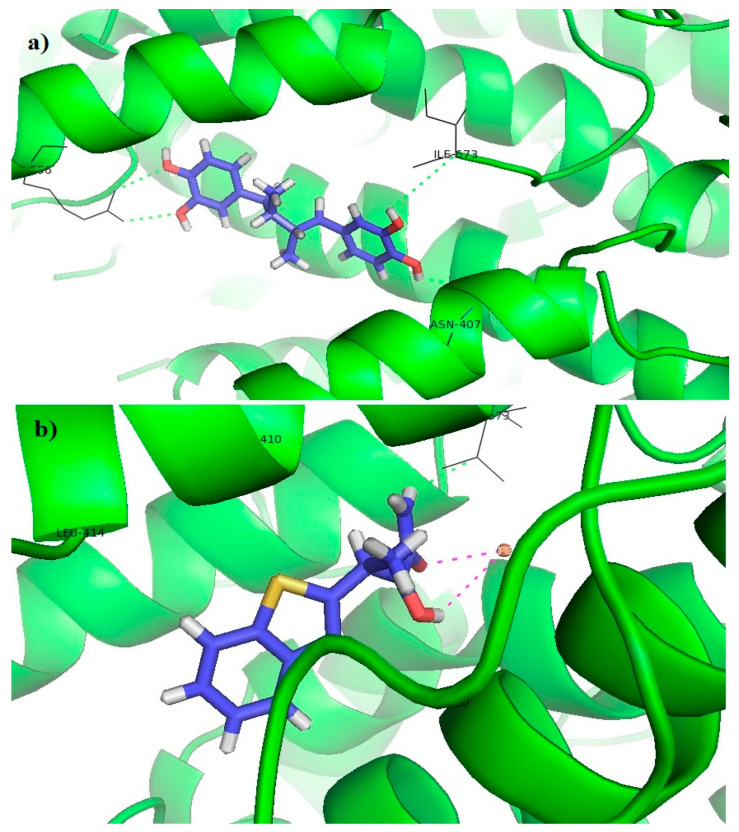
Molecular docking of (**a**) NDGA and (**b**) Zileuton into the 5-LOX enzyme.

**Figure 7 pharmaceutics-15-01441-f007:**
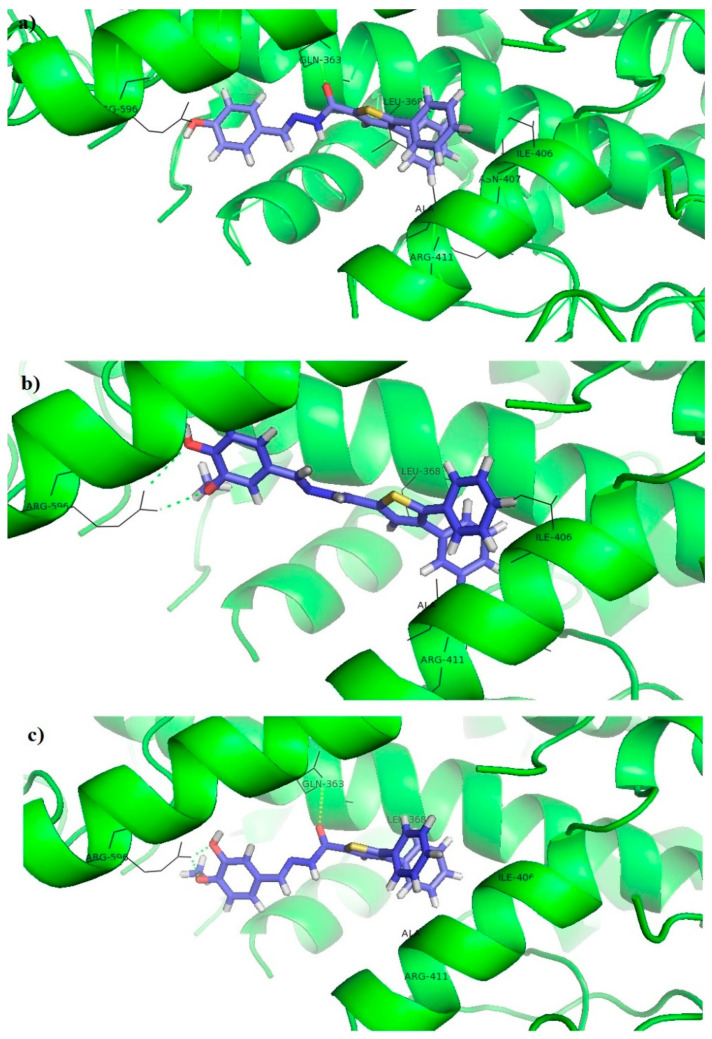
Molecular docking of the tested compounds into the 5-LOX enzyme: (**a**) **7c**, (**b**) **7e**, (**c**) **7f**.

**Table 1 pharmaceutics-15-01441-t001:** IC_50_ values of compounds in 2D cultured MIA PaCa-2 cell line.

Tested Compound	IC_50_ (µM)
**7a**	7.51 ± 0.08
**7b**	9.09 ± 1.11
**7c**	42.77 ± 3.59
**7d**	9.09 ± 1.04
**7e**	>100 µM
**7f**	4.86 ± 0.19
**7g**	24.23 ± 3.76
**7h**	>100 µM
**7i**	>100 µM
**7j**	9.65 ± 0.22
**DOX**	0.35 ± 0.18

DOX: Doxorubicin HCl; IC_50_: Inhibiting concentration of compounds that exhibited 50% cell viability.

**Table 2 pharmaceutics-15-01441-t002:** IC_50_ values of compounds in 2D cultured HaCaT cell line.

Tested Compound	IC_50_ (µM)
**7a**	0.88 ± 0.06
**7d**	>100 µM
**7f**	0.67 ± 0.13
**DOX**	0.11 ± 0.08

DOX: Doxorubicin HCl; IC_50_: Inhibiting concentration of compounds that exhibited 50% cell viability.

**Table 3 pharmaceutics-15-01441-t003:** Spheroid diameters of MIA PaCA cells treated with different concentrations of tested compounds.

Sample Concentrations		Spheroid Diameter (Square Millimeters)	
		Day 0	Day 9
Control	-	1111.8 ± 95.5	7240.2 ± 37.1
DOX	10 µM	1194.0 ± 63.3	910.0 ± 29.3
**7a**			
	0.1 µM	1156.5 ± 14.2	5050.5 ± 36.9
	1 µM	1186.5 ± 91.9	1755.0 ± 58.9
	10 µM	1176.0 ± 31.1	636.0 ± 12.7
	100 µM	1183.0 ± 36.1	432.0 ± 17.4
**7b**			
	0.1 µM	1193.0 ± 13.1	5922.5 ± 39.8
	1 µM	1185.5 ± 82.7	6461.5 ± 59.6
	10 µM	969.3 ± 28.9	4192.0 ± 44.8
	100 µM	644.7 ± 26.9	1050.5 ± 18.9
**7c**			
	0.1 µM	1118.0 ± 31.1	5415.0 ± 12.7
	1 µM	1168.0 ± 52.7	6174.0 ± 106.1
	10 µM	989.5 ± 54.1	3729.5 ± 12.8
	100 µM	940.5 ± 33.9	2207.0 ± 47.1
**7d**			
	0.1 µM	1127.5 ± 36.1	4911.0 ± 117.4
	1 µM	1211.0 ± 17.9	6312.5 ± 24.8
	10 µM	1297.5 ± 55.1	1708.0 ± 87.1
	100 µM	974.0 ± 32.2	445.7 ± 64.1
**7e**			
	0.1 µM	1181.5 ± 33.2	5607.5 ± 44.55
	1 µM	1153.0 ± 73.5	6396.0 ± 16.1
	10 µM	-	-
	100 µM	-	-
**7f**			
	0.1 µM	1110.0 ± 113.1	5823.0 ± 39.8
	1 µM	1108.0 ± 17.6	4336.0 ± 48.1
	10 µM	1215.0 ± 12.7	2806.0 ± 31.5
	100 µM	-	-
**7g**			
	0.1 µM	1222.5 ± 28.73	5969.0 ± 59.6
	1 µM	985.5 ± 65.7	3318.0 ± 35.4
	10 µM	-	-
	100 µM	-	-
**7h**			
	0.1 µM	1125.5 ± 28.9	5215 ± 18.3
	1 µM	1220.5 ± 31.5	6682.5 ± 7.1
	10 µM	1192.0 ± 24.7	8165.5 ±18.1
	100 µM	942.0 ± 58.9	7936.5 ± 48.6
**7i**			
	0.1 µM	1135.5 ± 2.5	5523.0 ± 15.2
	1 µM	943.0 ± 12.1	3539.0 ± 25.4
	10 µM	118.6 ± 24.7	2764.0 ± 37.9
	100 µM	1023.0 ± 59.5	2776.5 ± 85.2
**7j**			
	0.1 µM	1112.0 ± 35.4	6291.5 ± 45.87
	1 µM	998.0 ± 11.5	5934.0 ± 96.3
	10 µM	1068.0 ± 68.1	5217.4 ± 25.7
	100 µM	-	-

DOX: doxorubicin HCl.

**Table 4 pharmaceutics-15-01441-t004:** Results of the enzyme inhibition assays.

Compound	COX-2 IC_50_ (µM)	COX-2 Percent of İnhibition	COX-1 IC_50_ (µM)	5-LOX IC_50_ (µM)	5-LOX Percent of İnhibition
**7a**	-	13.57 ± 0.38	Not tested	3.78 ± 0.39	64.38 ± 0.46
**7b**	-	24.13 ± 1.55	Not tested	93.23 ± 9.89	25.77 ± 2.54
**7c**	10.13 ± 0.25	51.97 ± 2.51	˃100	2.60 ± 0.11	55.05 ± 1.37
**7d**	-	16.74 ± 3.36	Not tested	-	15.61 ± 0.66
**7e**	13.86 ± 0.76	49.41±1.12	˃100	3.30 ± 0.07	60.69 ± 0.54
**7f**	39.05 ± 3.41	38.41 ± 0.85	˃100	2.94 ± 0.20	60.56 ± 2.49
**7g**	-	11.41 ± 3.40	Not tested	-	20.38 ± 1.98
**7h**	-	8.35 ± 0.93	Not tested	-	19.87 ± 1.62
**7i**	-	13.27 ± 0.73	Not tested	-	27.23 ± 0.64
**7j**	-	16.64 ± 1.11	Not tested	-	36.10 ± 3.92
**Celecoxib**	0.07 ± 0.01	85.02 ± 0.31	˃100	Not tested	Not tested
**Zileuton**	Not tested	Not tested	Not tested	0.36 ± 0.10	64.85 ± 0.39

## Data Availability

All the data is available at Appendix A.

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
