# Peer review of "Discovery of Novel Thiophene/Hydrazones: In Vitro and In Silico Studies against Pancreatic Cancer"

_pharmaceutics, 2023, doi:10.3390/pharmaceutics15051441_

Round 1

Reviewer 1 Report

1.   Manuscript “Discovery of novel thiophene/hydrazones: in vitro and in silico 2 studies against pancreatic cancer” of Goknil Pelin Coskun et al. reports a study conducted on ten diarylthiophene-2-carbohydrazide derivatives. These new compounds were synthesized and evaluated for their utility on pancreatic cancer. They were tested on PaCa-2 cells and screened for their COX-2 and 5-LOX inhibitory activity. The molecular docking was also performed. These preliminary results suggested a potential as promising anticancer agents.

The work is original, well designed, and promising results are described in a relevant manner.

In my opinion, the work can be accepted with minor revision, as indicated below.

1.     The number of the compounds have to be written in bold everywhere

2.     Line 3: correct the small “s”

3.     Line 32; insert the ; among all the key words

4.     Lines 46-47: rewrite the sentence paying attention to punctuation

5.     Line 55: verify the word “specimens”

6.     Lines 59-61: rewrite and correct the sentence

7.     Line 91: indicate the eluents for TLC

8.     Line 93: also 13C?

9.     Line 95: use “multiplet”

10.  Line 95: indicate the use of ppm

11.  Lines 110-112: please improve the description of the synthesis introducing the eq. or the mmoL of the reagents and the amount of solvent (mL). Useful indicate the eluents.

12.  Line 114, figure 1: improve the quality, the table is too small. In the caption, the i-v have to be in the same style of the figure

13.  Line 117: (E) in italic (E) in all the IUPAC names

14.  Line 118: use “m.p.” everywhere

15.  Line 122: Indicate “ppm” at the end of all the list in H-and C-NMR

16.  Line 285, chemistry: please indicate the figure with scheme of synthesis in the text

17.  Line 302: indicate ppm of hydrazide protons used as reference

18.  Lines 312-313: indicate the kind of used cells in this experiment

19.  Line 316: explain the meaning of DOX as in the caption

20.  Line 322, paragraph 3.3: expand the SAR of this study

21.  Line 359: use “the IC50”

22.  Line 359: why do you used the COX1 at 10 uM? explain in the text this choice

23.  Line 353, paragraph 3.4: please, stress the results and better explain the SAR

24.  Lines 455-456: rewrite the sentence

25.  Line 451, conclusion: improve the final consideration and refer to the SAR

26.  Line 476: format the references as the instruction for authors

Author Response

Dear Reviewers;

Thank you for your time and effort for the review of our study. Your comments are highly important and we believe that the requested revisions will make the study even better for our readers. We have carefully checked and studied your suggestions. In order for the editorial and reviewers to follow the changes in the manuscript, we used ‘track changes’ in the Word document as the editor suggested. The reviewer response file prepared as ‘point by point’ including answers for all the suggestions. The answers were written in ‘red’ color for the reviewers to follow up the corrections.

In the name of all the authors, I personly thank you for your valuable contributions to our study.

Best Regards,

Reviewer 2 Report

The manuscript "Discovery of novel thiophene/hydrazones: in vitro and in silico studies against pancreatic cancer" shows new compounds with anticancer activity. However, authors should review the following points:

- When in vitro studies are carried out, we speak of non-anticancer antitumor activity (in vivo studies). Modify title, results, discussions and conclusions.

- Because MIA PaCa-2 cells (epithelial cell carcinoma) were chosen, and not pancreatic ductal adenocarcinoma, (a type of exocrine pancreatic cancer), which develops from the cells that line the small tubes, called ducts, in the pancreas.

- Studies must be carried out on non-tumor cell lines, to determine if the concentrations obtained are cytotoxic.

The text has moderate English. Authors should review the results and discussions sections.

Author Response

(The authors gave the same response as above.)
